

# Identification and validation of a novel signature for prediction the prognosis and immunotherapy benefit in bladder cancer

Yichi Zhang[1,2,*], Yifeng Lin[1,3,*], Daojun Lv[1], Xiangkun Wu[1], Wenjie Li[1], Xueqing Wang[4] and Dongmei Jiang[5]

[1] Department of Urology, the First Affiliated Hospital of Guangzhou Medical University, Guangzhou, China
[2] Nanshan School, Guangzhou Medical University, Guangzhou, Guangdong, China
[3] Department of Urology, Meizhou Hospital of Traditional Chinese Medicine, Meizhou, China
[4] Department of Ultrasound, Shantou Central Hospital, Shantou, Guangdong, China
[5] Department of Pathology, the First Affiliated Hospital of Guangzhou Medical University, Guangzhou, Guangzhou, China
* These authors contributed equally to this work.

Corresponding authors
Xueqing Wang,
m13414039585_1@163.com
Dongmei Jiang, 730245768@qq.com

## ABSTRACT

**Background:** Bladder cancer (BC) is a common urinary tract system tumor with high recurrence rate and different populations show distinct response to immunotherapy. Novel biomarkers that can accurately predict prognosis and therapeutic responses are urgently needed. Here, we aim to identify a novel prognostic and therapeutic responses immune-related gene signature of BC through a comprehensive bioinformatics analysis.

**Methods:** The robust rank aggregation was conducted to integrate differently expressed genes (DEGs) in datasets of the Cancer Genome Atlas (TCGA) and the gene expression omnibus (GEO). Lasso and Cox regression analyses were performed to formulate a novel mRNA signature that could predict prognosis of BC patients. Subsequently, the prognostic value and predictive value of the signature was validated with two independent cohorts GSE13507 and IMvigor210. Finally, quantitative Real-time PCR (qRT-PCR) analysis was conducted to determine the expression of mRNAs in BC cell lines (UM-UC-3, EJ-1, SW780 and T24).

**Results:** We built a signature comprised the eight mRNAs: CNKSR1, COPZ2, CXorf57, FASN, PCOLCE2, RGS1, SPINT1 and TPST1. Our prognostic signature could be used to stratify BC population into two risk groups with distinct immune profile and responsiveness to immunotherapy. The results of qRT-PCR demonstrated that the eight mRNAs exhibited different expression levels in BC cell lines.

**Conclusion:** Our study constructed a convenient and reliable 8-mRNA gene signature, which might provide prognostic prediction and aid treatment decision making of BC patients in clinical practice.

## INTRODUCTION

Bladder cancer (BC) is the most common diagnosed malignancy of the urinary tract system (*Torre et al., 2012*). Each year, bladder cancer is diagnosed in about 430,000 patients and is associated with approximately 165,000 deaths worldwide, making it one of the most lethal cancers (*Kamat et al., 2016*). Approximately 75% of patients with bladder cancers are non-muscle-invasive (NMIBC) at diagnosis and ~25% are muscle-invasive (MIBC). For BC patients who are at advanced stage (local progression or distant metastasis), cisplatin plus gemcitabine is regarded as the gold standard treatment. However, the anti-tumor effect is not satisfactory due to its low response rate and long-term therapeutic resistance (*Kaufman et al., 2000*). Currently, with the rapid development of immune checkpoint inhibitors (ICIs) treatment, such as cytotoxic T-lymphocyte antigen 4 (*CTLA-4*) and programmed cell death molecule 1 (*PD-1*)/ programmed cell death molecule ligand 1 (*PD-L1*) inhibitors, ICIs are replacing traditional therapeutic drugs and becoming new first-line and second-line treatment options for BC, marking the huge potential and hope of immunotherapy in BC. The median overall survival of BC patients receiving immunotherapy or chemotherapy was 10.3 months *vs.* 7.4 months (*Bellmunt et al., 2017*). Although ICIs has better efficacy compared with traditional platinum-based chemotherapy, it is estimated that only one fifth of solid tumor patients benefit from the treatment (*Fares et al., 2019*). Moreover, the overall efficacy of ICIs therapies remains unpredictable due to individual heterogeneity of genetics and immune microenvironment alterations as well as multiple confounders (*e.g.* lifestyle, metabolic disorders and sociological factors) (*Deshpande, Sharma & Watabe, 2020*; *Desrichard et al., 2018*; *Barr et al., 2016*). Hence, novel biomarkers that can accurately predict prognosis and therapeutic responses are urgently needed.

Accumulating evidences have confirmed a series of biomarkers including *PD-L1* expression, CD8$^+$ T cell, tumor mutational burden (TMB), and microsatellite instability (MSI) could act as biomarkers to predict clinical outcome and therapeutic responses in BC (*Patel & Kurzrock, 2015*; *Chan et al., 2019*; *Liu et al., 2020*; *Necchi et al., 2020*). However, these biomarkers seem to be insufficient for the therapeutic options and thus unable to be applied in clinical practice. What's more, incorporating molecular features and clinical information of BC patients into prediction model will provide better prediction effects (*Kim et al., 2011*; *Song et al., 2019*). Well-validated markers that predict survival benefits and immunotherapy efficacy were still an unmet need in BC.

Owing to advances in high-throughput sequencing, gene signatures at the mRNA level show great potential for predicting patient prognosis. Using transcriptomic profiles from 18 datasets, *Kamoun et al. (2020)* had successfully assigned MIBC patients into six molecular subtypes, in which responses to the treatment regimens may extremely vary. Similarly, in this study we used robust rank aggregation algorithm (RRA) to integrate differently expressed genes in five Gene Expression Omnibus (GEO) datasets. Lasso Cox regression was performed to develop a novel prognostic eight-gene immune signature and the stability and reproducibility was explored in independent datasets. The molecular mechanism and immune landscape relevance of the gene signature and prediction of the

potential response to immune checkpoint blockade was investigated. We also validated the expression levels of the eight genes in our clinical samples and cell lines. By applying this gene signature, we could accurately discriminate prognosis in a BC population and tailored the precision immunotherapy in BC patients.

## MATERIALS AND METHODS

### Study design and data collection

Data were collected as previously described in Zhang et al. (2019). Specifically, the inclusion criteria of GEO datasets were as follows: (1) biospecimens were gained from patients with localized BC; (2) enrolling at least 5 pairs samples in each dataset; (3) only including transcriptomic data in each dataset. (4) containing both clinical features (clinical tumor stage (TNM) or molecular subtype) and survival outcomes (OS or PFS).

The exclusion criteria were as follows: (1) duplicates of the previous eligible datasets; (2) not histo-pathologically confirmed urothelial carcinoma. Gene expression profile data (GSE37815, GSE13507, GSE121711, GSE40355, and GSE3167) were downloaded from the public Gene Expression Omnibus database (GEO, http://www.ncbi.nlm.nih.gov/geo/). Corresponding clinical information for GSE13507 was also obtained. Annotation information for the datasets and the platforms is shown in Table S1. The level 3 RNA-sequencing (RNA-seq) data (Fragments per kilobase million, FPKM) with the corresponding clinical information from 430 BLCA (Bladder Urothelial Carcinoma) samples (19 normal samples and 411 tumor samples) were downloaded from The Cancer Genome Atlas dataset (TCGA) (https://portal.gdc.cancer.gov/). The FPKM value was converted to the value per million transcripts (TPM) to make RNA-seq data more comparable with microarray data. The ENSEMBL IDs in RNA-seq data and the probes in microarray data were converted to gene symbol IDs using the annotation files from GENCODE (https://www.gencodegenes.org/). The IMvigor210 cohort was downloaded from http://research-pub.gene.com/IMvigor210CoreBiologies/#transcriptome-wide-gene-expression-data. It was a cohort evaluating the effect of atezolizumab (*PD-L1* blockade) in patients with locally advanced or metastatic urothelial BC. A flowchart of the analysis performed in this study is shown in Fig. 1.

### Data processing and screening for differentially expressed genes

To identify DEGs in the GEO datasets, the raw microarray data was converted to transcripts per million. The robust rank aggregation algorithm (RRA) in the "affy" package was used for background adjustment, log2 transformation and normalization. Then the RRA method was conducted to integrate the multiple-rank gene list of the five GEO datasets. The "edgeR" package was used to screen the DEGs from the TCGA dataset ($|\log2FC| > 1$, adjusted $P < 0.05$). Subsequently, we obtained the intersection of the DEGs from the two datasets by using Venn diagram tool (http://genevenn.sourceforge.net/).

### Identification and estimation of the prognostic multi-gene signature

After obtaining the DEGs in the previous step, univariate Cox regression analysis was conducted to determine which gene was significantly correlated with patients' OS

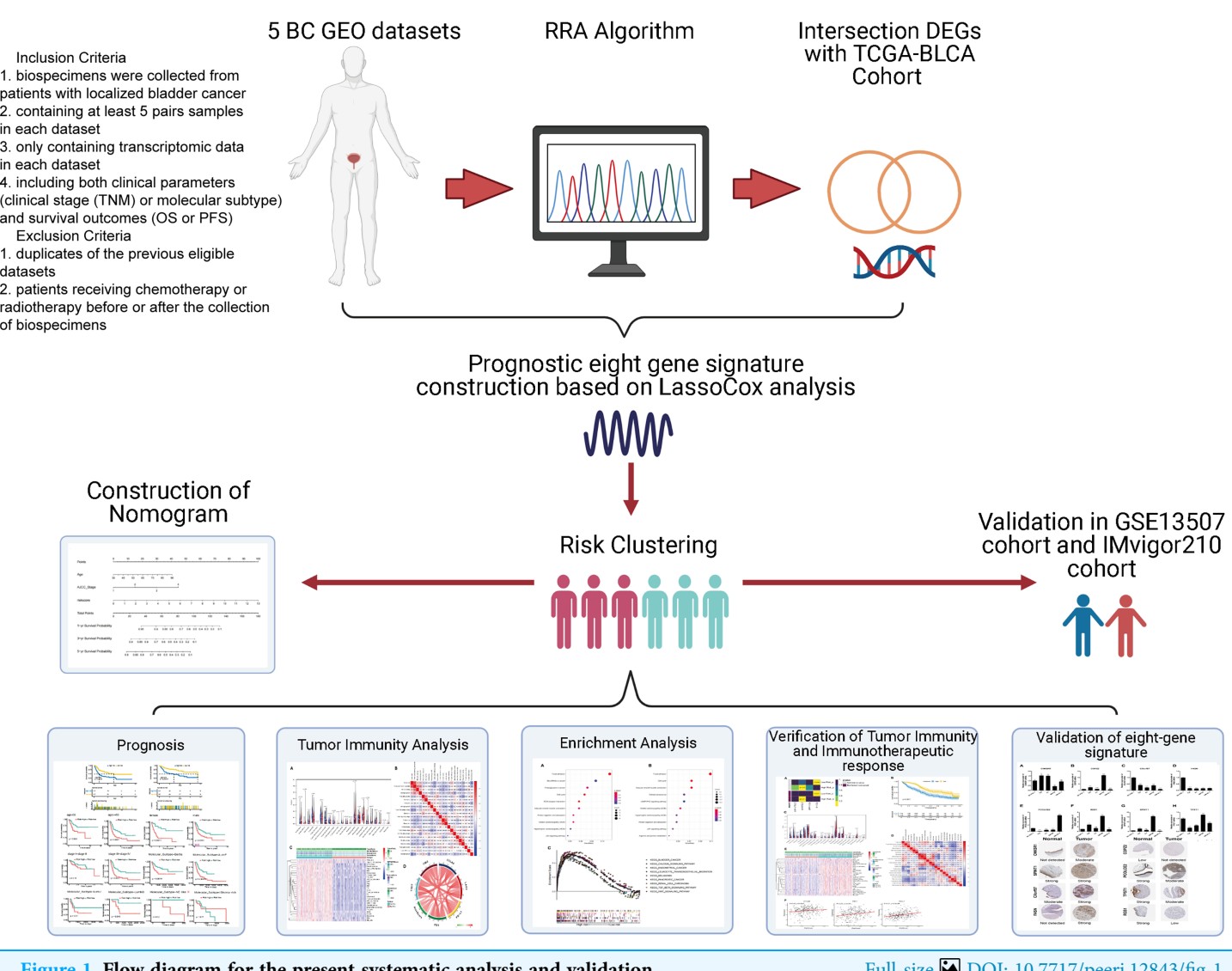

**Figure 1 Flow diagram for the present systematic analysis and validation.**

($P < 0.05$). Lasso-penalized Cox regression analysis was employed to further minimize the numbers of DEGs. The role of Lasso is to add a constraint condition to the sum of the absolute values of the coefficients, in order to reduce data dimensionality and interference and obtain better fitting. For high-throughput gene expression data with high-dimensional latitude and strong correlations, the Lasso method is practical after compressing meaningless explanatory variables to zero (*Gui & Li, 2005*). The regression coefficients of eight prognosis genes were derived from the stepwise multivariate Cox regression model, and then were used to calculate risk scores. All samples in the training (TCGA-BLCA) sets were divided into high- or low-risk groups by the cutoff values calculated by X-tile software. The Kaplan-Meier survival curves were performed to compare the survival risk between the two groups. Receiver operating characteristic (ROC) curves were conducted to demonstrate the predictive accuracy of this signature.

To determine the predictive power of the prognostic model and other clinical parameters including age, gender, T-stage, N-stage, M-stage, molecular subtype and pathologic stage, univariate and multivariate Cox analyses were performed. The information of six molecular subtypes (luminal papillary, luminal non-specified, luminal unstable, stroma-rich, basal/squamous, and neuroendocrine-like) were obtained from the Supplemental Material of the article (Kamoun et al., 2020). Parameters with $P < 0.05$ in the univariate analysis were incorporated to conduct the multivariate analysis. Kaplan–Meier analysis was performed to determine whether the effect value was consistent among different subgroups.

### Validation of prognostic multi-gene signature

In order to validate the predictive capability of this prognostic model, the GSE13507 database was used for external validation. The risk scores were established using the prognostic gene signature. Patients were grouped by the median risk scores using the same method as above. Kaplan–Meier analysis and ROC analysis were generated to evaluate the power of the model. The protein expression levels of the eight genes in normal and BC tissues were validated in the Human Protein Atlas database (https://www.proteinatlas.org/).

### Construction and validation of predictive nomogram

The independent clinicopathological parameters identified in multivariate analysis in the previous step were selected to develop a nomogram to predict the 1-, 3-, and 5-year overall survival (OS) of BLCA patients. The AUC of the ROC curve and concordance index were constructed to evaluate the predictive power of this prognostic model, and calibration plot was conducted to compare the accuracy of nomogram-predicted probabilities with actual observation. Decision curve analysis (DCA) was also used to determine the clinical utility of the nomogram model.

### Functional enrichment analyses

To reveal underlying functions of the prognostic gene signature in BC, GO and KEGG analyses were used with a false discovery rate (FDR) < 0.05 considered statistically significant. Patients from TCGA cohort were classed into two groups based on the risk scores as described previously. GSEA was performed to explore the potential molecular mechanisms enriched in the gene signature (FDR < 5%, nominal $P < 1\%$ and $|NES| > 1$).

### Tumor immunity analyses

To systematically illustrate the relationship between immune infiltrating cell phenotype and BC survival, we applied the CIBERSORT algorithm and assessed the relative proportions of 22 distinct leukocyte subsets (Newman et al., 2015). Only the CIBERSORT samples with $P < 0.05$ were filtered and chosen for further analysis. After filtering the data, the relevant violin plot and correlation heatmap were displayed by R package. The correlation between immune infiltration level and expression of each gene in the gene signature was explored through TIMER2.0 (http://timer.comp-genomics.org/). Since the accuracy for estimating the proportion of cell components were different by applying for different algorithm methods, we used QUANTISEQ algorithm to estimate the changes

in the proportion of CD8+ T cell and Tregs, and EPIC algorithm was used to estimate the changes in the proportion of CD4+ T cell, B cell, NK cell, macrophage, cancer associated fibroblast and endothelial cell (*Sturm et al., 2019*). To further explore the different infiltration degrees of immune cell types, immune-related functions, and immune-related pathways in two risk groups, single sample gene set enrichment analysis (ssGSEA) was applied by using the R package "GSVA" (*Hänzelmann, Castelo & Guinney, 2013*). The correlation between gene signature riskscore and immune related molecules was further investigated to better understand immune infiltration in BLCA.

## Prediction and evaluation of immunotherapeutic response

Tumor Immune Dysfunction and Exclusion (TIDE) algorithm were performed to predict the potential response to immune checkpoint blockade. Moreover, an independent cohort (IMvigor210) that recorded expression data from patients who responded or did not respond to anti-PD-L1 immunotherapy was used for further verification. Kaplan–Meier analysis was generated to test the prognostic value of gene signature. CIBERSORT and ssGSEA were applied to explore the immune landscape between high- and low-risk group stratified by risk score.

## Cell lines and culture conditions

The human BC cell lines (EJ, T24, UM-UC-3, SW780) and a normal human urinary tract epithelial cell line (SV-HUC-1) were purchased from Stem Cell Bank, Chinese Academy of Sciences (Shanghai, China). SV-HUC-1 was maintained in F-12K medium (Cat#11320033; Gibco, Grand Island, NY, USA). UM-UC-3, EJ and SW780 cell lines were routinely cultured in RPMI 1640 medium (Cat#11320033; Gibco, Grand Island, NY, USA), while T24 cell lines was cultivated in DMEM medium (Cat#11965092; Gibco, Grand Island, NY, USA). All the media were supplemented with 10% Fetal Bovine Serum (FBS, Gibco, Cat#10099141) and 1% penicillin/streptomycin (Cat#15070063; Gibco, Grand Island, NY, USA). All cells were incubated at 37 °C with in an atmosphere of 5% $CO_2$.

## Validation of key genes by the quantitative real-time PCR (qRT-PCR) analysis

Total RNA was extracted from cultured cell lines using TRIzol reagent (Cat# 9108; TaKaRa, Dalian, China). The ratio of absorbance at 260 and 280 nm (the A260/280 ratio) was used to evaluate the purity of RNA. Subsequently, cDNAs were synthesized by using the PrimeScrip$^{TM}$ RT reagent Kit (Cat# RR047A; TaKaRa, Dalian, China) from 1 µg of total RNA in 20 µl of reaction volume. 2×TB$^®$ Green qPCR Master Mix (Cat# RR820Q; TaKaRa, Dalian, China) was used for qRT-PCR with Roche Lightcycler 480 RT-PCR System. GAPDH fragment was used as an internal control for normalization of the data before calculation using the $2^{-\Delta\Delta Ct}$ method. Three independent replicates were conducted for each experiment. The primers used are listed in Table S2.

## Statistical analysis

All statistical analyses were conducted using the R software (version 3.6.2; https://www.r-project.org/). Cox regression model with Lasso based on the R package "glmnet" were

performed to generate optimal prognostic signature for BC. The Risk score was calculated using this formula: The risk score = $\sum_{i=1}^{n}(Coefi * Expi)$, where Expi represents the expression level of gene, i and coefi represents the regression coefficient of gene i in the signature. Kaplan–Meier plots and log-rank tests were used to compare the difference in the survival status between the high- and low- risk groups. The time-dependent receiver operating characteristic curve (ROC) and calculated the area under the curve (AUC) for 1-year, 3-year, and 5-year OS were conducted to validated the prediction ability of the risk signature. Mann–Whitney-Wilcoxon Test or Student's t test was executed to compare the difference between defined groups for continuous variables. Categorical clinical variables were assessed using Chi-square or Fisher's exact tests. Differences among BC cell lines and urinary tract epithelial cell line were analyzed by one-way ANOVA, and the Dunnett test was used as the *post hoc* test. $P < 0.05$ was set as the cutoffs for statistical significance.

## RESULTS

### Identification of robust DEGs

The annotation information for the datasets and platforms was exhibited in Table S1. We identified 306, 780, 4,035, 452, and 2,038 DEGs between normal and tumor tissues in the GEO datasets GSE3167, GSE37815, GSE40355, GSE13507, and GSE121711, respectively. Volcano plots displayed the distribution of DEGs in each dataset (Figs. 2A–2E). Based on the results of robust rank aggregation method 403 DEGs were screened out, including 132 upregulated and 271 downregulated genes. The top 20 up-regulated and down-regulated DEGs were listed in Fig. 2F. A total of 2,568 DEGs were selected from the TCGA-BLCA dataset, including 1,415 up-regulated and 1,153 down-regulated genes. By using Venn diagram web tool we chose a total of 302 DEGs for subsequent analysis in the two cohorts.

### Identification and validation of eight-gene prognostic signature

Univariate Cox regression modeling revealed 74-OS related genes in BLCA patients ($P < 0.05$). In the training set (TCGA-BLCA), Lasso-penalized Cox analysis was applied to further analyze the mRNAs data, and 19 genes were identified. The aim of backward stepwise regression was to construct a minimum set of independent variables by including all variables and deleting one variable at a time, to test which was least statistically significant. Model with the highest determination coefficient was built on the remaining variables (*Kuhn & Johnson, 2013*). 8-prognostic-gene model was finally selected by stepwise regression analysis: *CNKSR, COPZ2, CXorf57, FASN, PCOLCE2, RGS1, SPINT1* and *TPST1*.

The risk scores were calculated for all patient according to the following formula: $(0.0289 \times \exp_{COPZ2}) - (0.0573 \times \exp_{CNKSR1}) - (0.0602 \times \exp_{CXorf57}) + (0.00678 \times \exp_{FASN}) + (0.0297 \times \exp_{PCOLCE2}) - (0.0353 \times \exp_{RGS1}) + (0.0032 \times \exp_{SPINT1}) + (0.04 \times \exp_{TPST1})$.

In the training dataset, 394 patients were separated into high- and low-risk groups using the median risk score calculated by X-tile software. The low-risk group showed significant better overall survival compared with the high-risk group ($P < 0.001$; Fig. 3A).

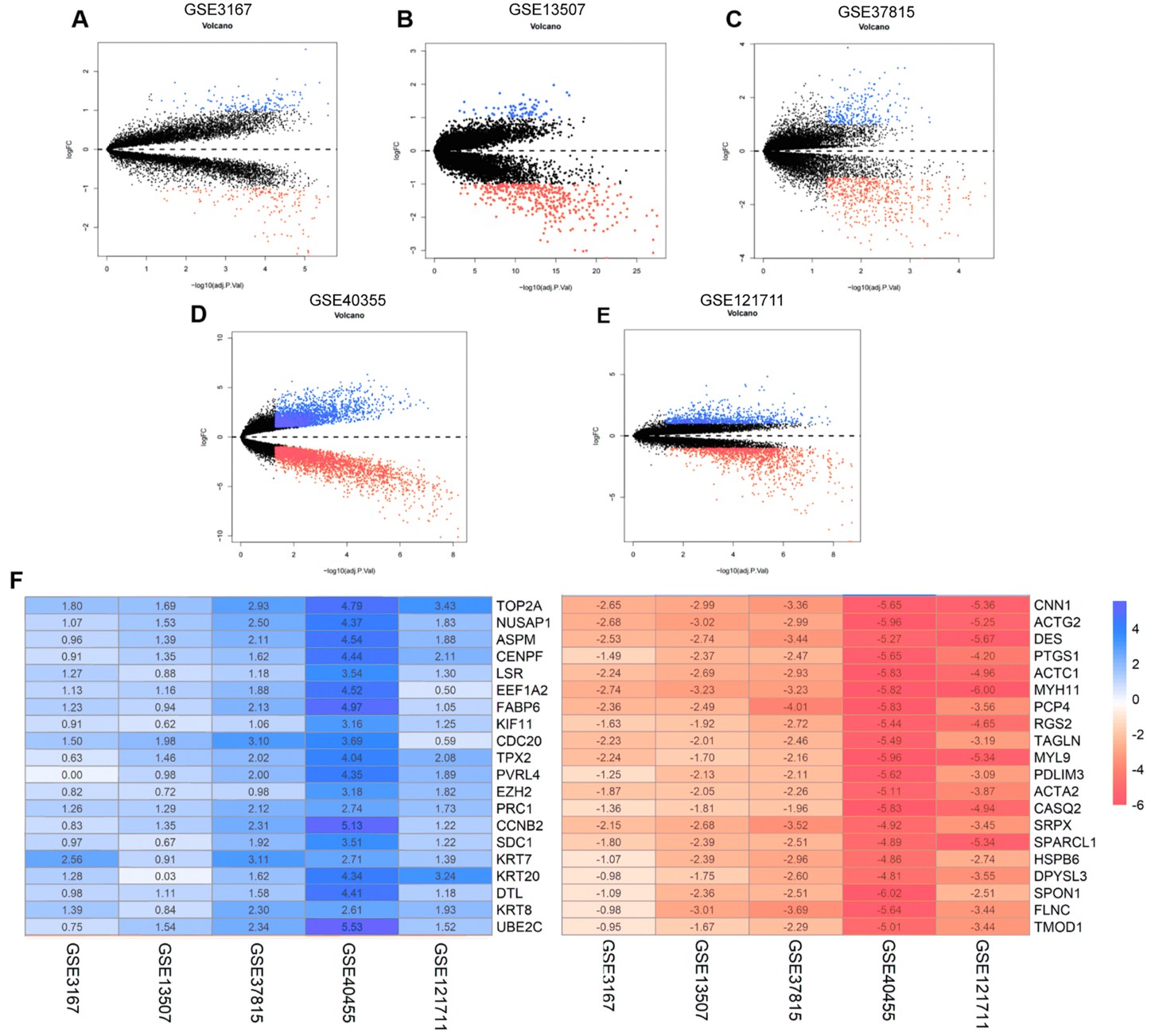

**Figure 2 The process of screening candidate genes.** (A) Volcano plot of DEGs in GSE3167. (B) Volcano plot of DEGs in GSE13507. (C) Volcano plot of DEGs in GSE37815. (D) Volcano plot of DEGs in GSE40355. (E) Volcano plot of DEGs in GSE121711. (F) Heatmap of top 20 upregulated and downregulated DEGs using robust rank aggregation analysis.

The AUC values of the ROC for 1-, 3-, and 5-year OS were 0.771, 0.735, and 0.718, respectively (Fig. 3C). Subgroup analyses stratified by age, gender, AJCC stage and molecular subtype were conducted to evaluate the prognostic values of the eight-gene signature in different subtypes. The patients with high riskscores had worse OS than the patients with low riskscores in age < 65 ($P < 0.0024$), age > 65 ($P < 0.0001$), female ($P = 0.00011$), male ($P < 0.0001$), stage III + IV ($P < 0.0001$), molecular subtype of

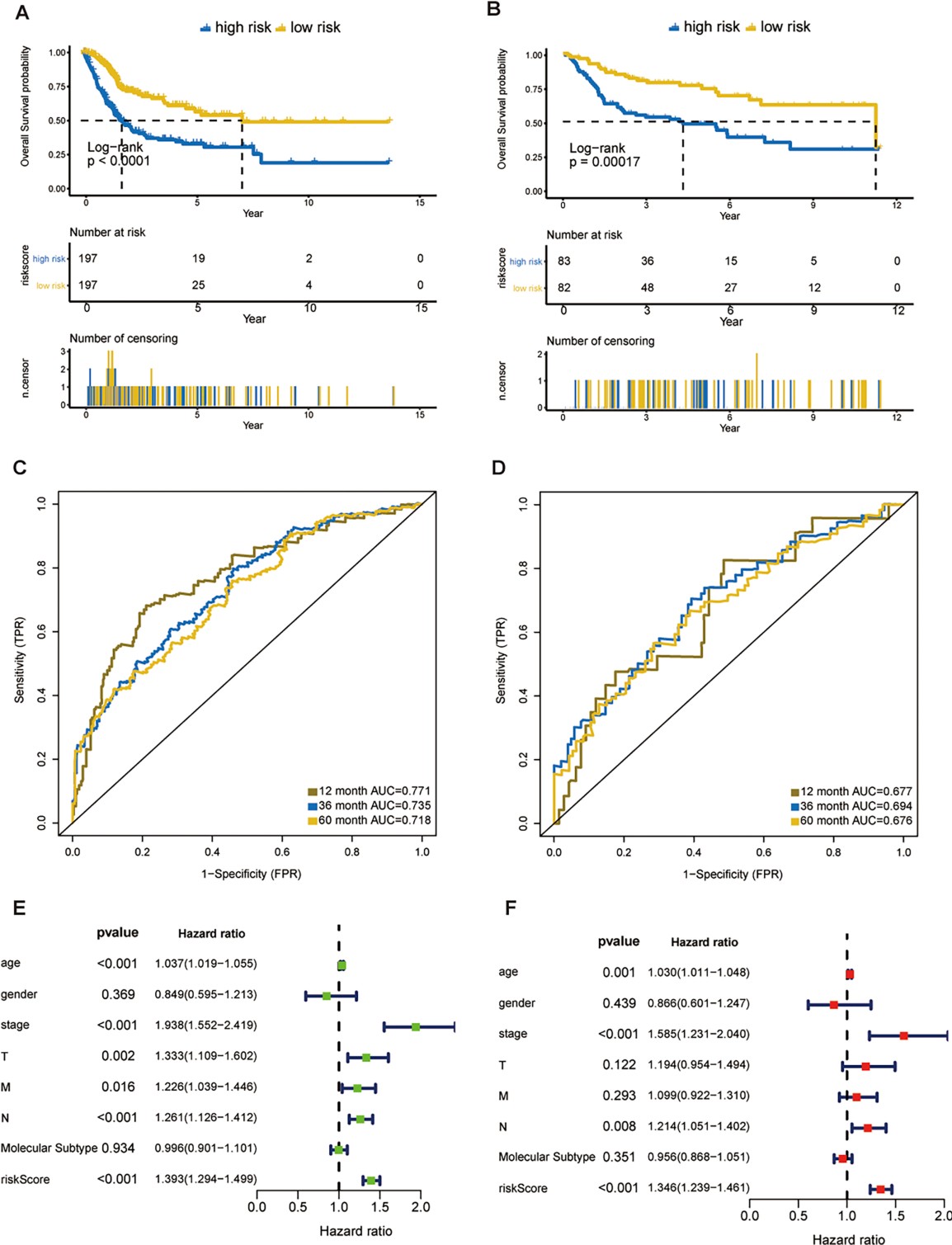

**Figure 3 Evaluation of the risk score formula in the training (TCGA-BLCA) and validation (GSE13507) sets.** (A) Survival analysis between signature-defined risk groups. Patients in the TCGA dataset were stratified into high- and low-risk groups according to the optimal cut-off values. (B) Time-dependent ROC curves for 1-, 3-, and 5-year OS predictions in TCGA dataset. (C) Survival analysis between signature-defined risk groups in GSE13507. (D) Time-dependent ROC curves for 1-, 3-, and 5-year OS predictions in GSE13507. (E and F) Univariable and multivariable analyses of riskscore, age, gender, pathologic stage, T/N/M stage and molecular subtype.               

basal/squamous ($P = 0.0019$), molecular subtype of luminal papillary ($P = 0.00017$) and molecular subtype of luminal non-specified ($P = 0.043$) (Fig. S1).

## Eight-gene signature is a prognostic factor independent of other clinicopathological parameters

In line with the results in the training dataset, the eight-gene signature could successfully stratify the samples in the validation dataset into different risk groups (Fig. 3B). The AUC values of the ROC for 1-, 3-, and 5-year OS were 0.677, 0.694, 0.676, respectively (Fig. 3D). Cox regression analyses were processed to access whether the eight-gene signature was an independent variable of BC patient survival. Univariate Cox regression revealed that 8-gene risk score, age, pathologic stage, T stage, and N stage were correlated with OS in BC (Fig. 3E). After the multivariate Cox analysis, risk score, age, and pathologic stage remained as independent prognostic parameters (Fig. 3F). The expression of *CNKSR1*, *CXorf57* and *FASN* at the protein level were significantly elevated in BC tissues compared with non-cancerous tissues, while the level of *COPZ2*, *PCOLCE2*, *TPST1* and *RGS1* were notably decreased in BC tissues than in normal tissues. No difference was found for *SPINT1* protein expression (Fig. S2). In addition, high risk group was correlated with a higher histological grade, T stage, N stage, M stage and clinical stage as shown in Table 1.

## Construction and validation of the predictive nomogram

In order to predict the prognosis of BLCA patients, we integrated the independent prognostic parameters, including 8-gene risk score, age and AJCC stage, to develop a nomogram model (Fig. 4A). The AUCs for the 1-, 3-, and 5-year OS were 0.769, 0.751, and 0.764, respectively (Fig. 4B). The C-index of the nomogram model was 0.703 (95% CI [0.547–0.872]), whereas that for AJCC stage was 0.634 (95% CI [0.451–0.818]), with 1,000 cycles of bootstrapping. Calibration plots showed that the results of predicted OS were consistent with the actual observations (Fig. 4C). DCA was used to evaluate the clinical utility of the nomogram model. The nomogram showed the greatest net benefit when compared with AJCC stage, T stage and risk score alone (Fig. 4D).

## GO/KEGG/GSEA

In biological processes, the gene signature was significantly enriched in focal adhesion, microRNAs in cancer, proteoglycans in cancer, cell cycle, and extracellular matrix receptor interaction; these processes have close connections with tumor proliferation and metastasis (Fig. 5A). The KEGG pathway analysis gave similar results (Fig. 5B).

To explore the significantly enriched pathways of the eight prognostic genes GSEA was performed. The results showed the high-risk group was enriched in more T cell suppressive pathways, such as the transforming growth factor β (TGF-β) signaling pathway, *Wnt* signaling pathway, and calcium signaling pathway (Fig. 5C).

## Correlation between tumor immunity and eight-gene signature

The association between risk score and the distribution of tumor-infiltrating immune cells was further investigated in two risk groups. Results showed that significantly higher

**Table 1 Clinicopathological characteristics of bladder cancer patients in the TCGA cohort.**

| Characteristics | All patients | Low risk | High risk | p-value |
|---|---|---|---|---|
| Patients, no. (%) | 392 (100) | 195 (49.7) | 197 (50.3) | |
| T stage, no. (%) | | | | 0.005 |
| T1+T2 | 116 (29.6) | 72 (36.9) | 44 (22.3) | |
| T3 | 189 (48.2) | 90 (46.2) | 99 (50.3) | |
| T4 | 57 (14.5) | 20 (10.3) | 37 (18.8) | |
| Unknown | 30 (7.7) | 13 (6.6) | 17 (8.6) | |
| N stage, no. (%) | | | | 0.016 |
| N0 | 227 (57.9) | 125 (64.1) | 102 (51.8) | |
| N1 | 125 (31.9) | 49 (25.1) | 76 (38.5) | |
| Unknown | 40 (10.2) | 21 (10.8) | 19 (9.7) | |
| M stage, no. (%) | | | | 0.024 |
| M0 | 187 (47.7) | 104 (53.4) | 83 (42.1) | |
| M1 | 10 (2.6) | 2 (1.0) | 8 (4.1) | |
| Unknown | 195 (49.7) | 89 (45.6) | 106 (53.8) | |
| Stage, no. (%) | | | | 0.002 |
| Stage I+II | 125 (31.9) | 76 (39.0) | 49 (24.9) | |
| Stage III | 137 (34.9) | 68 (34.9) | 69 (35.0) | |
| Stage IV | 130 (33.2) | 51 (26.1) | 79 (40.1) | |
| Grade, no. (%) | | | | <0.001 |
| Low Grade | 18 (4.6) | 18 (9.2) | 0 (0) | |
| High Grade | 372 (94.9) | 176 (90.3) | 196 (99.5) | |
| Unknown | 2 (0.5) | 1 (0.5) | 1 (0.5) | |
| Survival status, no. (%) | | | | <0.001 |
| Alive | 241 (61.5) | 146 (74.9) | 95 (48.2) | |
| Dead | 151 (38.5) | 49 (25.1) | 102 (51.8) | |
| Median months | 64.7 | 72.8 | 56.9 | |

proportions of M0 macrophages ($P = 0.013$), M2 macrophage ($P = 0.031$) and resting mast cells ($P = 0.04$) were exhibited in the high-risk group while the proportions of CD8$^+$ T cells ($P < 0.001$), CD4 memory-activated T cells ($P = 0.001$), regulatory T cells (Tregs) ($P = 0.003$) and follicular helper T cells ($P = 0.034$) were dramatically lower in the high-risk group (Figs. 6A and 6B). It's worth noting that the expression levels of eight genes were positively correlated with macrophage and cancer associated fibroblast infiltration (Fig. S3). BLCA samples were successfully divided into two clusters by applying "GSVA" algorithm. The ESTIMATE Score, Immune Score and Stromal Score were significantly higher in low-risk group than that of high-risk group. Also, low risk group had higher infiltration degrees of immune-related functions as well as immune-related pathways compared with high- risk group (Fig. 6C). Interestingly, the risk score was strongly positively correlated with the expression of immune checkpoints such as *PD-1, PD-L1, CTLA-4, TIM3, LAG3* and *TIGIT* (Fig. 6D).

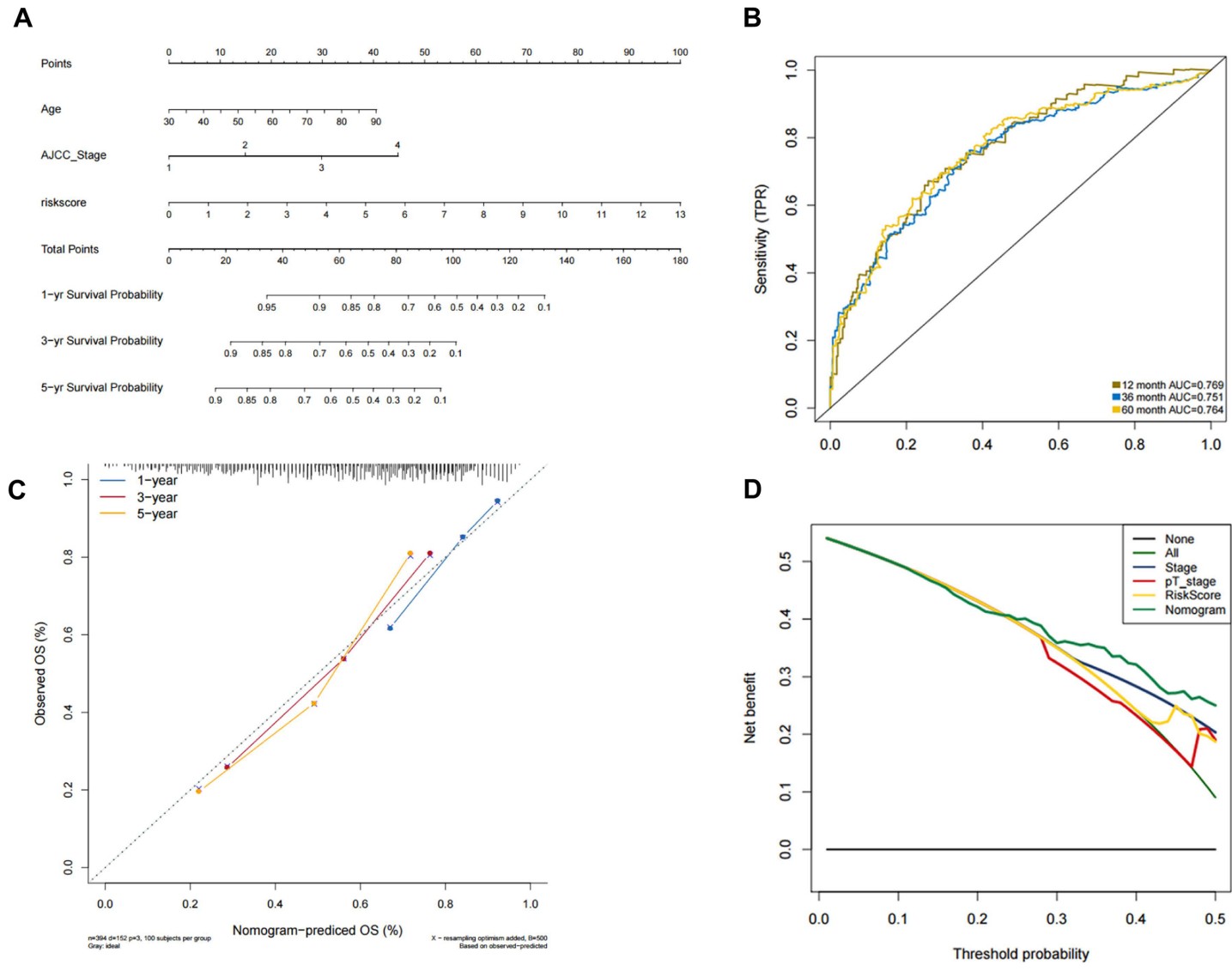

**Figure 4** **Validation of the nomogram for OS prediction in bladder cancer using the TCGA dataset.** (A) Nomogram predicting 1-, 3-, and 5-year OS. (B) Time-dependent ROC for 1-, 3-, and 5-year OS predictions using the nomogram. (C) Calibration plot of the nomogram. (D) DCA evaluation of the clinical utility of the nomogram.

## Verification of the correlation between the risk signature and immunotherapeutic response

The *CTLA-4* and *PD-1/PD-L1* blockade has emerged as a critical weapon in treating cancer (*Postow, Callahan & Wolchok, 2015*). Thus, we evaluated the eight-gene signature in predicting the sensitivity to ICBs in the two groups through TIDE algorithm and subclass mapping. The results showed that BLCA patients in low-risk group were more likely to response to *PD-1* blockade (Bonferroni corrected $P = 0.014$), and high-risk groups were more likely to response to *CTLA-4* blockade (Bonferroni corrected $P = 0.041$) (Fig. 7A). We investigated the prognostic value of the predictive model in IMvigor210

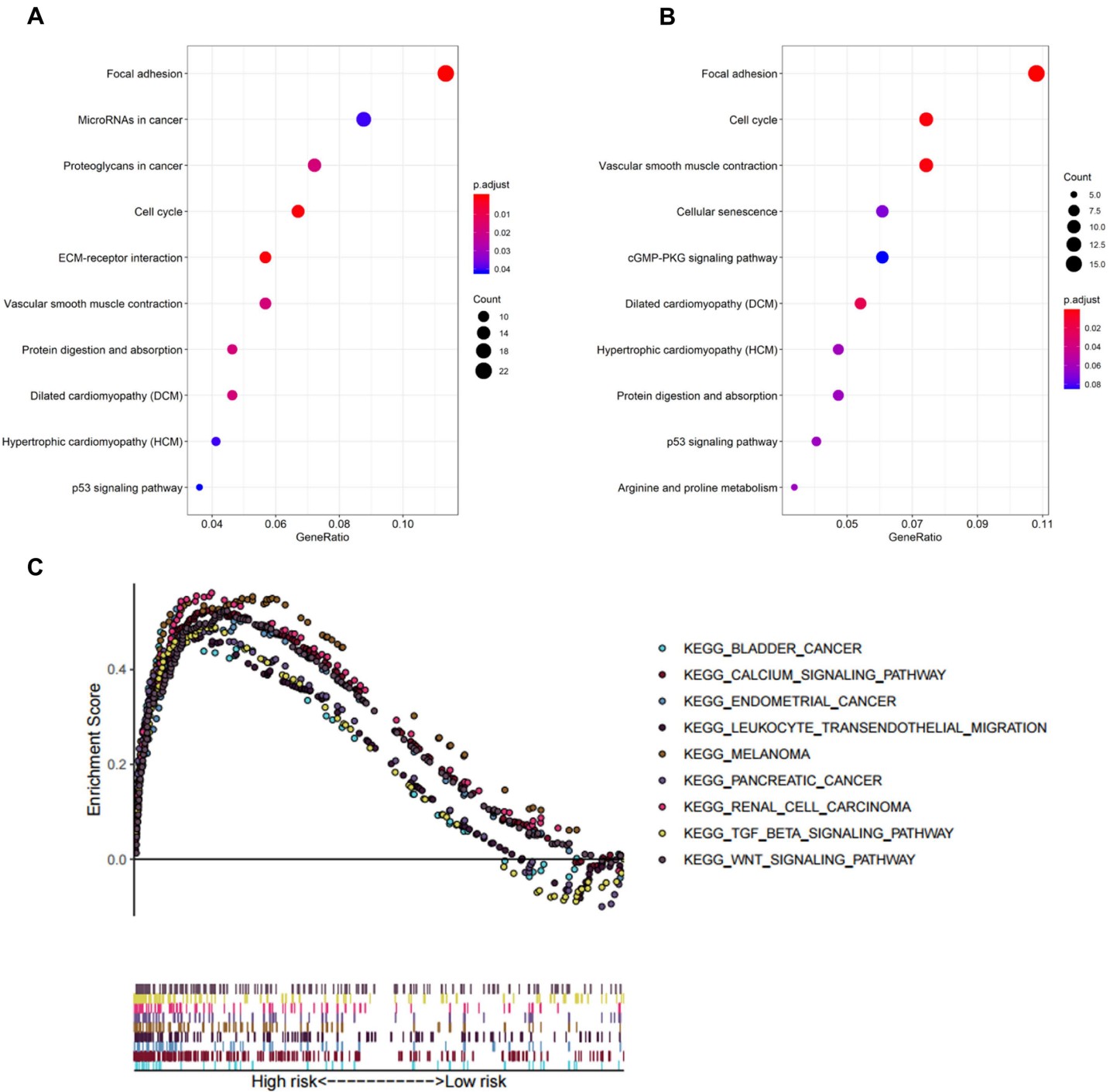

**Figure 5 Functional enrichment analysis of the eight-gene signature.** (A) GO enrichment analysis of the gene signature. (B) KEGG pathway enrichment analysis of the gene signature. (C) Biological processes enriched in the high-risk group using GSEA.

cohort, which included patients with metastatic urothelial cancer treated with anti-PD-L1 antibody. The results showed low-risk group stratified by the eight gene signature has a significant survival benefit compared with high-risk group (Fig. 7B). The validation

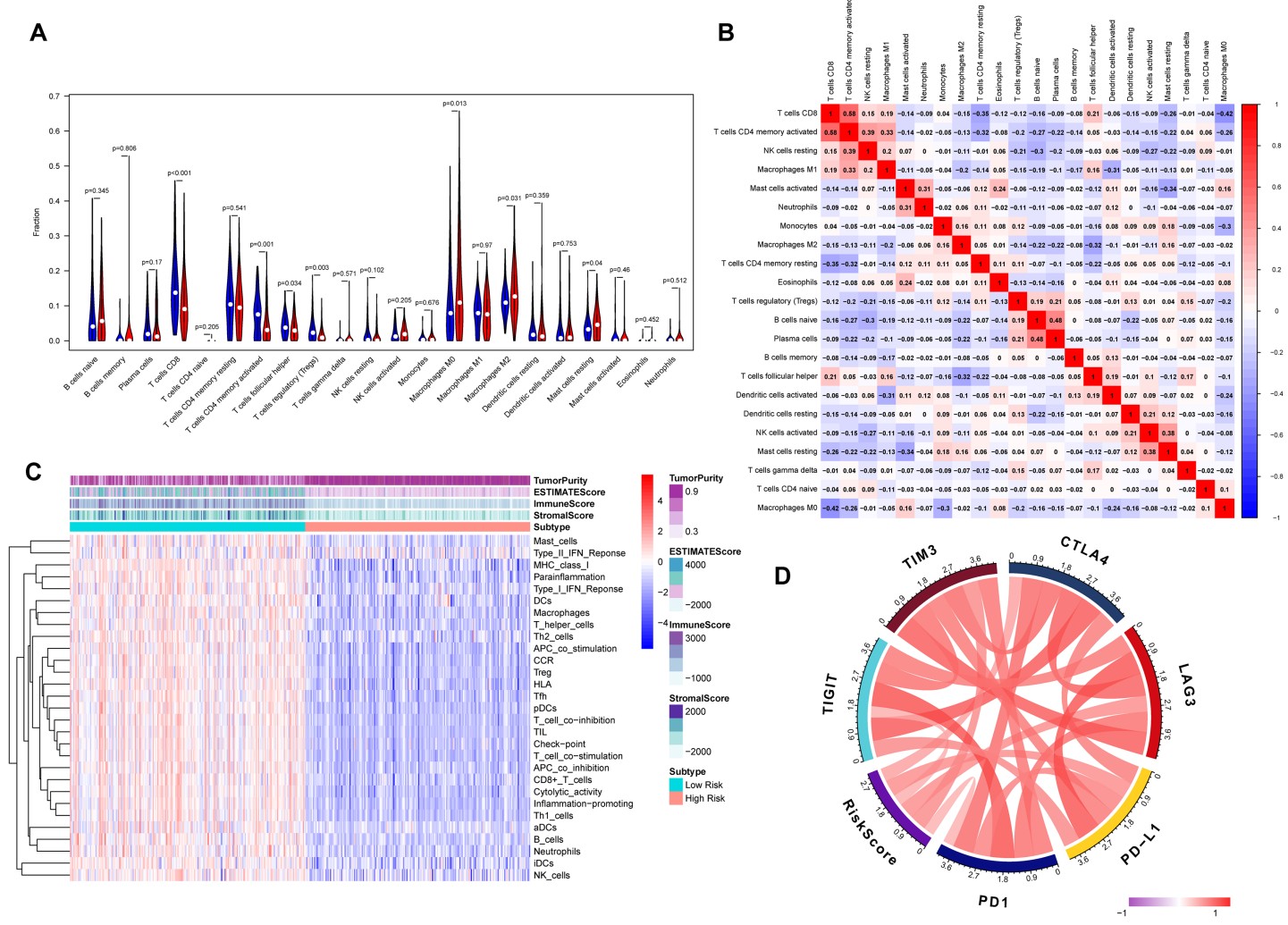

**Figure 6** **Tumor immune landscape based on the gene signature.** (A) The vioplot of 22 subpopulations of immune cells infiltration between high- and low-risk BC patients. (B) The corHeatmap for all 22 immune cell proportions. (C) ssGSEA analysis revealed the enrichment of immune-related functions, immune-related pathways and immune related cells between two risk groups. (D) Chord plot of the correlation between gene signature riskscore and immune related molecules.

cohort also showed a similar immune pattern with training cohort, and the risk score was found to be significantly correlated with immune-related molecules (Figs. 7C–7F).

## Distribution of risk score in different BC molecular subtypes

According to different transcriptomic and genomic profiling BC can be classified into six molecular subtypes, each with different clinical prognosis and therapeutic responses to chemotherapy and immunotherapy. We evaluated the distribution of risk score in six BC molecular subtypes, and the results showed the risk score was significantly different in different molecular subtypes (Fig. 8). Ba/Sq tumor with poor prognosis had the highest risk score, but the best prognostic outcome LumP and LumU had the lowest risk score.

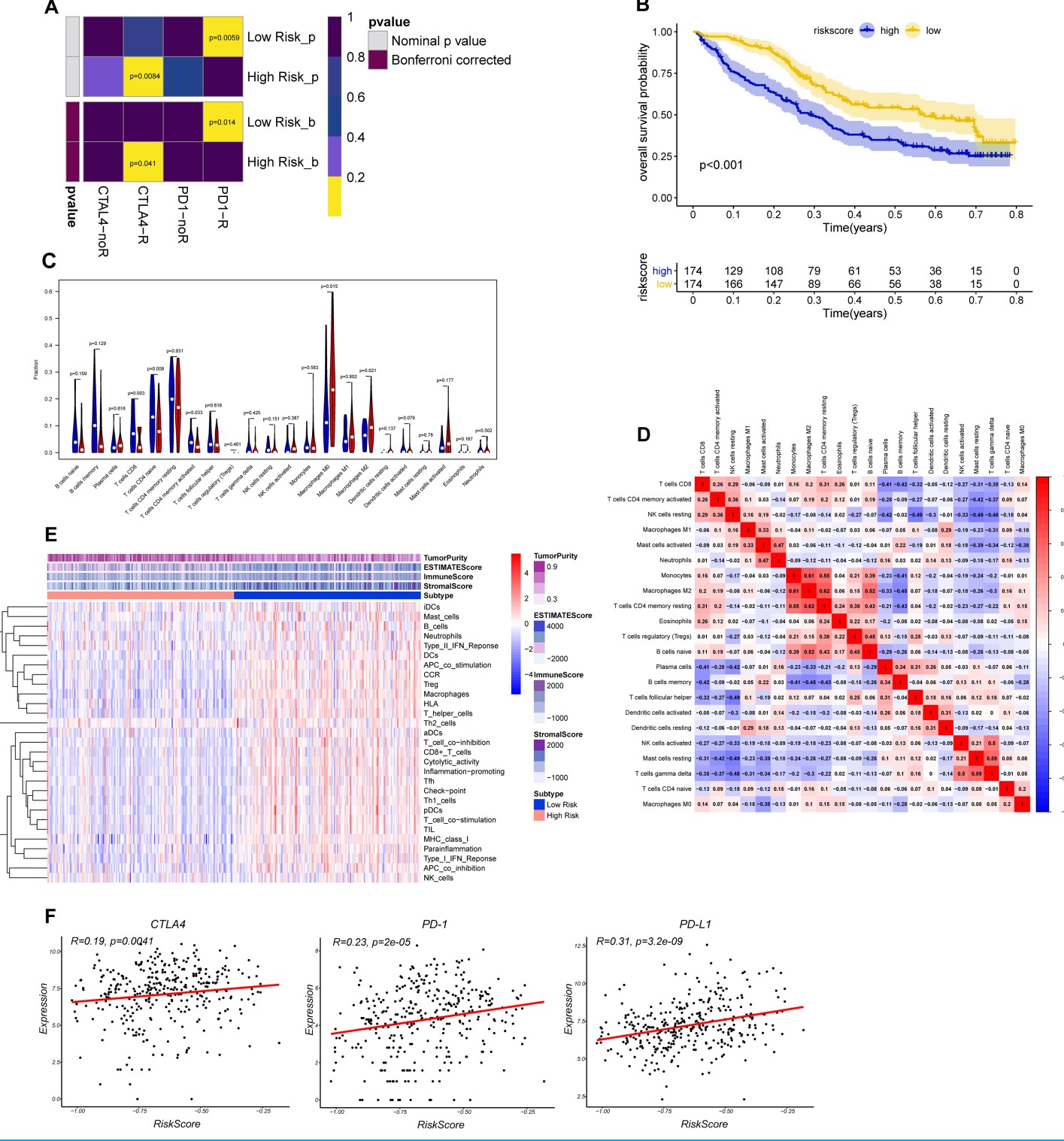

**Figure 7 Differential putative immunotherapeutic response prediction and verification of tumor immune landscape in IMvigor210 cohort.**
(A) Submap analysis of immunotherapeutic responses to anti-CTLA-4 and anti-PD-1 treatments in low- and high-risk group. (B) Survival analysis between signature-defined risk groups. Patients in IMvigor210 cohort were stratified into high- and low-risk groups according to the optimal cut-off values. (C) The violin plot of 22 subpopulations of immune cells infiltration between high- and low-risk groups in IMvigor210 cohort. (D) The corHeatmap for all 22 immune cell proportions. (E) Exploration of tumor microenvironment between two risk groups in IMvigor210 cohort using ssGSEA analysis (F) Correlation between gene signature risk score and immune checkpoint molecules.

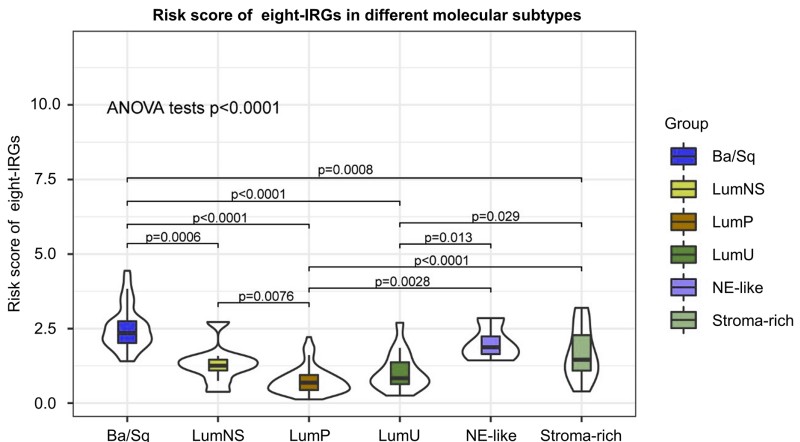

**Figure 8** **Risk score of novel eight-gene signature in six bladder cancer molecular subtypes.**

## Validation of the expression levels of the eight genes in cell lines

The expression signatures at the mRNA level of the 8 genes were then investigated in four human BC cell lines (*i.e.* EJ, T24, UM-UC-3 and SW780) and demonstrated that the mRNA levels of *CNKSR1* were relatively higher in most of the BC cells when compared with normal human urinary tract epithelial cell line (SV-HUC-1), however *CXorf57* and *FASN* were dramatically decreased in most of the BC cells (Figs. S4A–S4H).

## DISCUSSION

Considerable heterogeneity at the genomic, transcriptional, and cellular levels have been found in BC, which may result in diverse clinical characteristics and contribute to different responses to chemotherapy, immunotherapy, and targeted therapies (*Meeks et al., 2020*). Genetic biomarkers have identifiable molecular characteristics that can be used to detect disease, evaluate prognosis, and monitor tumor progression or therapeutic response (*Khailany et al., 2020*). Previous reports have demonstrated promising gene-based risk signature models that enable immune monitoring of tumors and individualized treatment (*Qu et al., 2018*; *Lu et al., 2020*; *Wang et al., 2021*). As a new treatment method, ICIs exert a favorable effect in patients with BC. However, some patients show unresponsiveness to ICIs treatment. We hope to identify a prognostic gene signature which can not only predict the prognosis of BC patients but also select the potential patients that may show favorable response to ICIs.

In this study, a panel of eight-gene signature comprising *CNKSR1, CXorf57, FASN, SPINT1, COPZ2, PCOLCE2, TPST1*, and *RGS1* was finally selected to generate a risk score model which can be exploited for predicting survival in BC. The predictive performance of the signature was mutually verified in internal TCGA-BLCA and external GSE13507 dataset. Survival analysis revealed that high-risk group exhibited significantly worse prognosis than low-risk group. A nomogram integrated with both the 8-gene-based signature and clinicopathological risk factors demonstrated that the model can accurately predict patients' overall survival (OS). The AUC, C-index, DCA, and calibration curves all

indicated the good predictive performance of this model. The risk score showed a powerful ability to differentiate BLCA into different risk groups and together with the nomogram, could facilitate the clinical use in clinical practice.

Interestingly, these eight genes which were used to establish our risk signature model have been shown to be involved in tumor development and thus may be promising therapeutic targets for BC. *FASN* has previously been reported as an oncogene in many cancer types (*Menendez & Lupu, 2007*). Its product is the only human protein to catalyze *de novo* synthesized long-chain fatty acids, implying FASN plays a crucial role in lipid metabolism in tumor microenvironment. The upregulation of FASN enhances lipogenesis in tumor cells, mainly through activation of the AKT/mTOR signaling pathway, thereby contributing to the immunometabolic switch in the tumor microenvironment and enhancing tumor growth and proliferation. In addition, metformin, a commonly used drug for treating diabetes, has exerted potent inhibitory effect in tumor growth through targeting SREBP-1c and its downstream target FASN, thus inhibiting lipogenesis in bladder cancer (*Deng et al., 2021*). Studies have shown that blockade of FASN exerts a novel effect in inhibiting growth of BC cells (*Tao et al., 2019*; *Zheng et al., 2016*). *SPINT1* (*HAI-1*, hepatocyte growth factor activator inhibitor type 1), is an inhibitor of transmembrane serine protease that regulates matriptase activity. Hepatocyte growth factor (*HGF*) contribute to tumor development through its specific receptor *MET* (*Tervonen et al., 2016*; *Kawaguchi & Kataoka, 2014*). Most fibroblasts secrete the inactive single-chain precursor (pro-*HGF*) and matriptase is one of the most potent activators of pro-*HGF* (*Lee, Dickson & Lin, 2000*). *HAI-1* suppresses matriptase-mediated conversion of pro-*HGF* into its active form (*Ye et al., 2019*). *Shimwell et al. (2013)* in their study found a significantly elevated concentration of *HAI-1* in the urine sample and identified *HAI-1* as potential urinary biomarker through the combination of transcriptomics and proteomic analyses. *HAI-1* may also be involved in cell migration and bladder cancer cell metastasis (*Chen et al., 2021*). *Yamasaki et al. (2018)* revealed that low expression of *HAI-1* was related to poor prognosis in BC. However, this was inconsistent with our results. Further experiments are needed to elucidate the mechanism by which *HAI-1* participates in the development of BC. *CXorf57*, also known as *RADX*, is a single-stranded DNA-binding protein. By modulating the activity of *RAD51*, which is known to be involved in the homologous recombination and repair of DNA, it promotes replication fork stability (*Schubert et al., 2017*; *Dungrawala et al., 2017*). *CNKSR1* acts as a scaffold component for receptor tyrosine kinase signaling and is involved in many signal transduction pathways, including *PI3K-Akt*, *Ras*, *MAPK*, and *NF-κB* signaling (*Fritz, Varga & Radziwill, 2010*; *Fritz & Radziwill, 2010*; *Indarte et al., 2019*; *Farhan et al., 2010*; *Fischer et al., 2016*). Overexpression of *CNKSR1* may promote proliferation and invasion in human breast and cervical cancer (*Fritz, Varga & Radziwill, 2010*; *Fritz & Radziwill, 2010*). However, other studies have shown that high *CNKSR1* expression was associated with less aggressive biological characteristics of pancreatic cancer, suggesting that it may be helpful in the selection of patients for surgical resection (*Quadri et al., 2017*). *Wang et al. (2021)* found that CNKSR1 protein was mainly expressed in the cytoplasm and the expression of CNKSR1 protein was significantly higher in MIBC than in normal

bladder samples. However, its underlying role in remaining bladder tumor growth remained unclear. *COPZ1* and *COPZ2* are two isoforms encoding the ζ subunit of coat protein complex 1, which functions as a vesicle carrier in secretory pathways (*Beck et al., 2009*). Downregulation of miR-152 and its host gene *COPZ2*, together with upregulation of *COPZ1* led to suppression of autophagy and apoptosis in cancer cells (*Shtutman et al., 2011*; *Tsuruta et al., 2011*). *PCOLCE2* encodes a glycoprotein, procollagen COOH-terminal proteinase enhancer (*PCPE*), which regulates collagen fibril deposition in the extracellular matrix and plays an important role in remodeling the tumor microenvironment (*Steiglitz, Keene & Greenspan, 2002*). *PCOLCE2* is regarded as a biomarker for gynecological cancers, squamous cell carcinoma of head and neck and non-small-cell lung cancer (*Tian, Meng & Zhang, 2019*; *Lim et al., 2017*; *Zhang & Wang, 2019*). Consistent with our results, low *PCOLCE2* expression is linked to better OS. Tyrosine sulfation, which is catalyzed by tyrosyl protein sulfotransferase 1 (*TPST1*) and *TPST2*, leads to changes in protein-tyrosine sulfotransferase, transferase, and protein homodimerization activity. *TPST1* is overexpressed in nasopharyngeal carcinoma and is associated with tumor invasion (*Zhao et al., 2004*). Expression levels of *TPST1* in lung cancer tissue are significantly lower than those in normal tissue and are inversely association with the expression of *c-Met* (*Jiang et al., 2015*). However, the biological function of *TPST1* in BC remains unclear. *RGS1* is a GTPase-activating protein and it can regulate the function of G-proteins (*Hollinger & Hepler, 2002*). Studies have shown RGS1 can desensitize chemokine receptor signaling and cause decreased chemotaxis of lymphocytes in lymphoid organs (*Moratz, Harrison & Kehrl, 2004*). We imputed *RGS1* may lead to lymphocyte dysfunction and contribute to immune escape in BC.

To uncover the molecular mechanisms of the gene signature and their potential biological functions, GO, KEGG, and GSEA enrichment analyses were executed. Based on the results of GO and KEGG analysis, we found that the eight genes may play important parts in tumor proliferation and metastasis. GSEA revealed enrichment of *TGF-β* and Wnt/β-catenin signaling in the high-risk group. *TGF-β* is an immunosuppressive factor that plays a crucial part in cancer development through promoting dysplasia, angiogenesis and epithelial-to-mesenchymal transition. Also, *TGF-β* may impair the anti-tumor T cell response (*Gabrilovich, 2004*). *Mariathasan et al. (2018)* revealed that blockade of *TGF-β* enhanced the potency of anti-*PD-L1* antibody (atezolizumab) by boosting the activity of CD8$^+$ T cells and hence enhancing anti-tumor immunity. In urothelial cancers key molecules that regulate the *Wnt/β-catenin* pathways are being used as diagnostic biomarkers and potential therapeutic targets (*Garg & Maurya, 2019*). *Spranger, Bao & Gajewski (2015)* found that aberrant *Wnt/β-catenin* activation led to defects in CD103$^+$ dendritic cells recruitment and a subsequent decrease in *CXCL9* and *CXCL10* secretion, thereby preventing T-cell infiltration. We also evaluated the correlation between tumor immunity and the eight gene signature both in internal TCGA-BLCA and external IMvigor210 cohort. Our results revealed that significantly higher levels of CD8$^+$ T cells, CD4$^+$ T cells and lower levels of macrophages were exhibited in low-risk group than in high-risk group. The expression levels of eight genes were positively correlated with macrophages and cancer associated fibroblast infiltration, and these cells played an

important role in the creation of immunosuppressive TME and had a negative prognostic effect on BC. In addition, risk score was also positively correlated with immune checkpoint molecules such as *PD-1, PD-L1* and *CTLA4*. We imputed that once these immune checkpoint molecules were activated, they may promote tumor cells and other "bad cells" to escape the immune system. The landscapes of immune infiltrating cells, immune-related function and immune-related pathways of the high- and low-risk groups indicated that the low-risk group tended to have inflamed tumors, whereas the high-risk group had excluded tumors (*Galon & Bruni, 2019*). Increasing evidences have indicated that immune inflamed subtypes which characterized by infiltration of $CD8^+$ T cells have optimal response to ICIs therapy (*Galon & Bruni, 2019*; *Kato et al., 2020*; *Keam et al., 2020*). Excluded tumors are a subtype characterized by retention of cytotoxic T cell in excessive reactive stroma but not the absence of T cells. Thus, this kind of tumor is still responsive to immune checkpoint inhibitors.

TIDE algorithm indicated that low-risk BLCA patients may be responsive to *PD-1* blockade therapy, and high-risk BLCA patients may be responsive to *CTLA4* blockade therapy. Based on the above results we imputed that ICIs may still yield survival benefits in high-risk BLCA patients but the normal anti-tumor immunity needed to be restored within tumor parenchyma (such as inhibition of fibroblasts in tumor stroma, elimination or transformation of tumor-associated macrophages (TAMs), Tregs and myeloid-derived suppressor cells (MDSCs), inhibition of *TGF-β* and Wnt/β-catenin signaling). Recent research on neutralization of TGF-β led to tumor stroma remodeling and enhanced the efficacy of immunotherapy also provide rationale for "targeted + immune" treatment in immune excluded tumors (*Grauel et al., 2020*). These results showed great effectiveness of our eight-gene-based risk signature in distributing BC patients into inflamed or excluded subtype, which may benefit from different ICIs based treatment. It might serve as a biomarker in tailoring individualized immunotherapy.

To our knowledge, the eight-gene signature presented here has not been previously reported and is more cost-effective and practical in clinical utility than whole-genome sequencing. Considering the intra- and inter-tumor heterogeneity of BC tumors we sought to find variable in transcriptional levels which are more suitable to explain the complex interplay between tumors and the immune system, when compared to models incorporated clinical parameters only. We used rank aggregation analysis and integrated five GEO and TCGA datasets to identify the final gene signature. Hence, our results are highly reliable and robust. As BC is a heterogeneous cancer and can be classified into different molecular subtypes, each with different clinical prognosis and therapeutic responses to chemotherapy and immunotherapy; and we found risk score was significantly different in different molecular subtypes. The results of RNA-sequence in cell lines demonstrated that aberrant expression levels of *CNKSR1, CXorf57* and *FASN* may promote proliferation and induce malignant transformation in BC. Moreover, a nomogram combining the eight-gene signature with clinicopathological parameters further improved the predictive power and the possibility of clinical use. Although the risk signature exerts a robust predictive value in risk stratification and guidance for treatment options, its accuracy and effectiveness need to be further confirmed in

substantial clinical trials. Also, *in vivo* studies investigating the role of the eight genes are needed to verify our results in the future.

## CONCLUSIONS

In summary, our study identified a novel signature that would be applied as a prognosticator and a promising biomarker in individualized immunotherapy for BC. These findings improve our understanding of immunotherapies in BC and provide valuable indication for future studies.

## ABBREVIATIONS

| | |
|---|---|
| **BC** | bladder cancer |
| **ICIs** | immune checkpoint inhibitors |
| **TME** | tumor microenvironment |
| **OS** | overall survival |
| **HR** | hazard ratio |
| **CI** | confidence interval |
| **AJCC** | American Joint Committee on Cancer |
| **DEGs** | differentially expressed genes |
| **AUC** | Area Under the Curve |
| **ROC** | Receiver-operator Characteristic curve |
| **Lasso** | least absolute shrinkage and selection operator |
| **GSEA** | gene set enrichment analysis |
| **GO** | gene ontology |
| **KEGG** | Kyoto Encyclopedia of Genes and Genomes |
| **TCGA-BLCA** | The Cancer Genome Atlas Bladder Urothelial Carcinoma |
| **GEO** | Gene Expression Omnibus |

### Funding

This study was supported by the Guangdong Basic and Applied Basic Research Foundation (No. 2019A1515110033); Distinguished Young Talents in Higher Education Foundation of Guangdong Province (No. 2019KQNCX115); China Postdoctoral Science Foundation (No. 2019M662865); Science and Technology Plan Project of Guangzhou (No. 202102010150 and 202102080010); Achievement Cultivation and Clinical Transformation Application Cultivation Projects of the First Affiliated Hospital of Guangzhou Medical University (No. ZH201908); Student Science and Technology Innovation Project of Guangzhou Medical University (No. 01-407-2101). The funders had no role in study design, data collection and analysis, decision to publish, or preparation of the manuscript.

## Grant Disclosures

The following grant information was disclosed by the authors:

Guangdong Basic and Applied Basic Research Foundation: 2019A1515110033.

Distinguished Young Talents in Higher Education Foundation of Guangdong Province: 2019KQNCX115.

China Postdoctoral Science Foundation: 2019M662865.

Science and Technology Plan Project of Guangzhou: 202102010150 and 202102080010.

First Affiliated Hospital of Guangzhou Medical University: ZH201908.

Guangzhou Medical University: 01-407-2101.

## Competing Interests

The authors declare that they have no competing interests.

## Author Contributions

- Yichi Zhang conceived and designed the experiments, performed the experiments, analyzed the data, prepared figures and/or tables, authored or reviewed drafts of the paper, and approved the final draft.
- Yifeng Lin analyzed the data, prepared figures and/or tables, and approved the final draft.
- Daojun Lv analyzed the data, prepared figures and/or tables, authored or reviewed drafts of the paper, and approved the final draft.
- Xiangkun Wu analyzed the data, authored or reviewed drafts of the paper, and approved the final draft.
- Wenjie Li performed the experiments, prepared figures and/or tables, and approved the final draft.
- Xueqing Wang analyzed the data, authored or reviewed drafts of the paper, and approved the final draft.
- Dongmei Jiang conceived and designed the experiments, authored or reviewed drafts of the paper, and approved the final draft.

## Data Availability

The dataset of TCGA-BLCA generated and/or analyzed during the current study are available at the TCGA database (https://cancergenome.nih.gov/).

The datasets generated and/or analyzed during the current study are available at GEO: GSE3167, GSE37815, GSE121711, GSE40355 and GSE13507.

The dataset of IMvigor210 generated and/or analyzed during the current study are available in http://research-pub.gene.com/IMvigor210CoreBiologies/#transcriptome-wide-gene-expression-data.

## Supplemental Information

Supplemental information for this article can be found online at http://dx.doi.org/10.7717/peerj.12843#supplemental-information.

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
