# Peer review of "Identification and validation of a novel signature for prediction the prognosis and immunotherapy benefit in bladder cancer"

_PeerJ, doi:10.7717/peerj.12843_

## Round 0.1 · original submission · Major Revisions

When assessing your paper, the reviewers identified some issues that must be addressed for the manuscript to be suitable for publication in this journal. Additionally, please address how your current study is different from a previously published study by Wang et al., Cancer Med, 2020 (https://www.ncbi.nlm.nih.gov/pmc/articles/PMC7571842/). The results and conclusions of your manuscript appear to be nearly identical to Wang et al publication.

Reviewer 2 has suggested that you cite specific references. You are welcome to add it/them if you believe they are relevant. However, you are not required to include these citations, and if you do not include them, this will not influence my decision.

Reviewer 1 ·

Basic reporting

1. There are numerous grammatical errors and typos that affect readability.
2. Some of the references are outdated and shall be replaced with more up-to-date literature.

Experimental design

1. The authors used five GEO datasets for the screening of differentially expressed genes between normal and tumor tissues. How are these datasets selected? What are the selection/exclusion criteria for datasets?
2. Have the authors compared the prediction performance with other methods? Does this 8-gene signature outperform other gene signatures in predicting prognosis and responsiveness to immunotherapy? Does it outperform models using only clinical variables, which will be much less expensive and laborious to conduct? (An example would be the six-factor prognostic model for patients with advanced urothelial carcinoma receiving post platinum atezolizumab by Pond et al., 2018)
3. RT-PCR analysis of mRNA expression of the genes in established cancer cell lines does not necessarily provide validation of the gene signature nor of the prediction model. First, the changed gene expression in the GEO or TCGA datasets might represent the upregulation or downregulation of genes in the tumor microenvironment (e.g., immune cells, tumor-associated fibroblasts, etc.), but not that in the tumor cells. Second, cancer cell lines can behave very differently from patient tumor samples because of culture conditions. Also, the authors used a cell line called “EJ”. This is confusing because there are several cell lines named EJ, including a human diffuse large B-cell lymphoma cell line.

Validity of the findings

It is important to develop prognostic models that can identify patients who might benefit most from new immunotherapies, Once confirmed, these tools could provide a critical decision-making tool for clinicians. However, previous literature has described numerous immune-responding signatures in bladder cancer. For example, Lv et al1. have reported a risk model based on eight immune-related genes that predict the prognosis of BLCA patients, which can also “predict response to immunotherapy, reflect different BLCA subtypes, immune and mutation status”. Comparably, Fu et al2. have built a 30 gene risk signature to predict the effect of immunotherapy in BC patients. Moreover, in an almost identical approach, Wang et al. have developed a 13-gene signature that can “classify the immunotherapeutic susceptibility of patients with BC”. The Result and Conclusion sections of the current manuscript are nearly identical to that of Wang et al., Cancer Med, 2020.

The authors need to compare their findings to previous work and explain explicitly how their “discovery” is different from previous reports and highlight their novelty.

Additional comments

NA

Annotated reviews are not available for download in order to protect the identity of reviewers who chose to remain anonymous.

Reviewer 2 ·

Basic reporting

no comment

Experimental design

no comment

Validity of the findings

no comment

Additional comments

In the present research, the authors identify the gene signature as a predictive indicator of immunotherapy treatment for BC.
I have several reservations. My comments are appended as below:

Major comments:
1. Abstract- In the background section, please specify the dearth of studies in the present literature and how your research fills the gap. In methods, please specify the cell lines used. Results- please specify the 8 gene-based signatures.
2. Introduction- Authors should first discuss the general facts on prognosis and pathological types of BC, especially with reference 2.
3. Introduction- please include median survival on checkpoint therapy in BC.
4. Introduction- authors should revisit a few reviews while discussing the cofounders for immunotherapy efficacy. For instance- PMID: 33076303, PMID: 31102620
5. Authors should describe the GEO dataset characteristics (may include a table with relevant information). Was it includes immunotherapy responders and non-responders? How do authors stratify the samples?
6. In the flowchart, please show inclusion and exclusion criteria.
7. Figure 2- please annotate the volcano plot with the corresponding GEO dataset. In addition, authors should provide the complete list of genes in supplemental tables.
8. I have a question on the figure 2F- the commonly predicted genes as PD-L1, Inf gamma signaling was not visible in the list. Authors should justify.
9. Line 216- please elaborate the regression analysis. Which datasets were used?
10. Figure 3 A, B- is it PFS or OS?
11. Validation of nomogram- authors should use responder/nonresponse cohort. It is not clear the characteristics of the samples used.
12. It is not clear to me the rationale to use the cell lines. Are they tested to be sensitive/resistant for selected treatment?
Minor comments:
1. Please provide catalog number of all used reagents.
2. Line 179- qRT PCR- please specify how RNA purity was detected and what quantity of RNA was used.
3. Survival curved- please specify PFS/OS

Reviewer 3 ·

Basic reporting

The language used is very clear , and well supported with the correct references.

Just a few minor corrections are suggested:
1. In the introduction, on line 75, correct the spelling "gens" to genes.
2. In the discussion, on line 311, change the word "Previously" to Previous.

Experimental design

This study identified a novel signature ,that would be applied as a prognosticator and a promising biomarker in individualized immunotherapy for BC. The experimental design and the methods employed, sufficiently address the main objective of this study.

Validity of the findings

All the data provided, is well supported with correct rationale and is statistically sound.

One minor suggestion - Please elaborate further on the role of the 8 individual genes, in promoting Bladder cancer specifically.

Additional comments

A well written manuscript, with excellent findings , which can help design better individualized treatments for bladder cancer.

---

## Round 0.2 · accepted · Accept

Thank you for revising the manuscript as per the reviewers' recommendations.

Reviewer 1 ·

Basic reporting

No comment

Experimental design

No comment

Validity of the findings

No comment

Additional comments

The authors’ response letter and the revised manuscript have addressed most of my concerns. In particular, the authors have compared their gene signature with existing literature and showed that the novel 8-gene signature is comparable or superior to previously described gene signatures. The authors also moved their not-so-relevant RT-PCR results to the supplementary data. I have one remaining minor concern about figure 4. The figure, especially Fig 4C, is hard to read because of poor resolution and small font size. After this is addressed, I would recommend the manuscript for publication.

Reviewer 2 ·

Basic reporting

no comment

Experimental design

no comment

Validity of the findings

no comment

Additional comments

I congratulate the authors for providing the modifications. All my concerns are successfully addressed. I recommend accepting this manuscript.

Reviewer 3 ·

Basic reporting

The corrections have been made , as per suggestions. No further changes needed.

Experimental design

No further changes needed.

Validity of the findings

The justification provided for the queries in the discussion, have been responded to , appropriately.